# AEON: Adaptive Estimation of Instance-Dependent In-Distribution and Out-of-Distribution Label Noise for Robust Learning

## Abstract

Robust training with noisy labels is a critical challenge in classification because it offers the potential to reduce reliance on costly clean-label datasets. Real-world datasets often contain a mix of in-distribution (ID) and out-of-distribution (OOD) instance-dependent label noise, which is a challenge that is rarely addressed simultaneously by existing methods and is further compounded by the lack of comprehensive benchmarking datasets. Furthermore, many noisy-label learning methods rely on inefficient multistage learning algorithms that explicitly identify noisy samples, which are treated differently from those presumed to be clean. However, this identification typically depends on either oversimplified two-component clustering assumptions (clean vs. noisy) or the arbitrary specification of a noise rate. To address these research gaps, we propose the **A**daptive **E**stimation of Instance-Dependent In-Distribution and **O**ut-of-Distribution Label **N**oise (**AEON**) method. The AEON is an efficient one-stage noisy-label learning methodology that adaptively estimates instance-dependent (ID and OOD) label noise rates to enhance robustness in complex noise settings. In addition, we introduced a new benchmark that reflects real-world ID and OOD noise scenarios. Experiments demonstrate that AEON achieves state-of-the-art performance on both synthetic and real-world datasets[1].

## 1 Introduction

Traditionally, classification relies on meticulously curated datasets that are predominantly composed of clean-label samples (Wu et al., 2020; Zhang et al., 2024). However, real-world datasets often lack careful curation, resulting in a significant proportion of instance-dependent label noises. This noise typically appears in two forms: (1) **closed-set or in-distribution (ID) noise**, in which samples are incorrectly labeled but belong to one of the training categories, and (2) **open-set or out-of-distribution (OOD) noise**, in which samples are mislabeled with one of the training categories, but originate from categories outside the training label set (Fig. 1).

Recent studies revealed that real-world datasets commonly exhibit ID and OOD instance-dependent label noise (Northcutt et al., 2021; Beyer et al., 2022; Li et al., 2017; Xiao et al., 2015). However, existing methods generally address these noise types independently using inefficient multistage learning algorithms (Albert et al., 2023). For ID noise, many approaches include a learning stage that employs clustering methods (e.g., two-component Gaussian mixture models) on sample loss values to separate data into clean and noisy categories (Arazo et al., 2019; Li et al., 2020). For OOD noise, one of the learning stages typically employs energy-based scoring techniques to separate clean samples from noisy samples (Albert et al., 2022a; Liu et al., 2020b). This compartmentalized approach overlooks the coexistence of ID and OOD noise in real-world datasets, thereby limiting the applicability of current methods in practical scenarios. Addressing this research gap remains a critical yet scarcely explored challenge for robust classification.

Efforts to simultaneously address both ID and OOD label noise have emerged (Wei et al., 2021; Sachdeva et al., 2021; Albert et al., 2025; 2023), but these approaches often rely on ad-hoc parameters to reflect the respective noise rates (Li et al., 2020; Han et al., 2018; Yao et al., 2020). The

---

[1]Code will be open-sourced upon acceptance.

Figure 1: Different types of samples labeled as "Airplane": Clean-Set (●) has samples with correct labels, Closed-Set (●) contains samples with incorrect labels, where the image class ("Bird") is in the set of training labels, and Open-Set (●) has samples with incorrect labels, where the image class ("Helicopter") is not in the set of training labels.

accurate estimation of these noise rate parameters could significantly enhance their effectiveness; however, only a few methods have attempted this systematically so far. A notable example is (Garg et al., 2024), which estimates ID noise rates but does not attempt to jointly analyze ID and OOD label noise, leaving this critical gap largely unaddressed in the literature.

Furthermore, most existing methodologies conceptualize OOD noise as random corruption (Albert et al., 2025; 2023; 2022a), overlooking its inherently instance-dependent nature. Such simplification ignores the complexities of real-world datasets, where both ID and OOD noise often vary systematically across instances (Song et al., 2022; Garg et al., 2024). In addition, common synthetic benchmarks typically assume instance-independent OOD noise (Jiang et al., 2020), producing test scenarios that fail to capture the instance-dependent noise patterns prevalent in practice (Xia et al., 2020; Yao et al., 2021b).

This study addresses the critical challenges of instance-dependent ID and OOD label noise, together with the new **A**daptive **E**stimation of Instance-Dependent In-Distribution and **O**ut-of-Distribution Label **N**oise (**AEON**) method. The AEON consists of an efficient one-stage learning algorithm that simultaneously estimates instance-dependent ID and OOD label noise rates, thereby enabling robust learning in scenarios with complex ID and OOD label-noise patterns. In addition, we propose a new Instance-Dependent Combined Open- and Closed-Set Noise (ID2Noise) benchmark to better capture the challenges of real-world datasets. In summary, our contributions are as follows:

- The novel AEON method, an efficient one-stage noisy-label learning framework that jointly estimates open-set and closed-set noise rates to effectively address instance-dependent ID and OOD label noise.

- The new ID2Noise benchmark to systematically evaluate methods for learning with noisy labels in simulated instance-dependent ID and OOD noise scenarios.

Our method achieves accurate ID and OOD noise rate estimation while being more computationally efficient than existing SOTA methods. On standard benchmarks, such as CIFAR-100 (Krizhevsky & Hinton, 2009) with mixed noise (40% ID and 40% OOD), our approach delivers a performance gain of $\approx 3\%$ in accuracy compared with state-of-the-art (SOTA) approaches. More critically, on our challenging ID2Noise benchmark, we achieve $\approx 47\%$ accuracy, whereas competing methods fall below $\approx 38\%$, highlighting the importance of not only AEON, but also of the proposed ID2Noise benchmark to assess robust noisy-label learning methods.

## 2 PRELIMINARIES

### 2.1 PROBLEM FORMULATION

Let $\hat{\mathcal{X}} \subseteq \mathbb{R}^d$ denote the input space and $\hat{\mathcal{Y}}$ represent a $C$-dimensional one-hot vector space defined by $\hat{\mathcal{Y}} = \{\hat{y} : \hat{y} \in \{0,1\}^C \wedge \mathbf{1}_C^\top \hat{y} = 1\}$, where $\mathbf{1}_C$ denotes a $C$-dimensional vector of 1s. Given a noisy-label training dataset $\mathcal{D} = \{(\hat{x}_i, \hat{y}_i)\}_{i=1}^N$, where $\hat{x}_i \in \hat{\mathcal{X}}$ represents the input data and $\hat{y}_i \in \hat{\mathcal{Y}}$ represents the corresponding noisy labels, our goal is to learn a robust classifier $f_\theta : \hat{\mathcal{X}} \to \mathbb{R}^C$ parameterized by $\theta$ with a robust learning method that leverages the estimation of the ID and OOD noise rates to accurately predict clean labels, denoted as $y$.

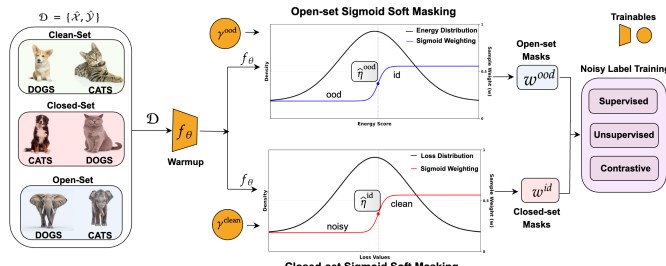

Figure 2: Our proposed AEON is a novel one-stage learning framework to simultaneously address instance-dependent ID and OOD label noise. The framework comprises three key components: (1) a warm-up phase that establishes the initial feature representations through $f_\theta(.)$; (2) A dual-stream soft-masking mechanism that dynamically estimates sample reliability through energy scores (for OOD label noise detection) and loss values (for ID label noise identification), producing adaptive weights $w^{\text{ood}}$ and $w^{\text{id}}$, respectively, via noise rate estimation ($\hat{\eta}^{\text{id}}$ for the closed set, and $\hat{\eta}^{\text{ood}}$ for the open set); and (3) a unified multi-objective training strategy combining supervised learning on clean-label ID samples, unsupervised learning on potentially noisy-label ID instances, and contrastive learning for robust feature discrimination on OOD samples.

## 2.2 SAMPLE SELECTION IN MULTISTAGE LNL METHODS

Existing multistage LNL methods (Han et al., 2018; Li et al., 2020; Kim et al., 2021; Liu et al., 2020b; Hao & Li, 2024) usually have one stage to automatically select samples that have clean and noisy labels, and another stage for training the model using separate loss functions for each set of samples. This selection of samples requires a function $h : \hat{\mathcal{X}} \to \mathbb{R}$ to characterize samples in terms of their likelihood of having clean or noisy labels. Sample separation is often performed by either *(i)* clustering the values of the criterion evaluated on all training samples into two clusters (e.g., 2-component Gaussian mixture model), or *(ii)* thresholding via an arbitrary threshold value. The separation process results in two groups of samples: clean labels and noisy labels.

For ID noise, methods such as Co-teaching (Han et al., 2018) or DivideMix (Li et al., 2020) rely on a criterion based on the loss function evaluated on each sample (also known as the small loss criterion):

$$h(\hat{x}_i) = \ell_{\text{CE}} \left( f_\theta(\hat{x}_i), \hat{y}_i \right) = -\hat{y}_i^\top \log \left\{ \text{softmax} \left[ f_\theta(\hat{x}_i) \right] \right\}. \tag{1}$$

These loss values are then clustered via a 2-component Gaussian mixture model, and samples belonging to the cluster with the smaller mean value are considered clean, whereas others are considered noisy. In FINE (Kim et al., 2021), the criterion is the dot product between the largest eigenvector of the class-specific Gram matrix and the sample itself, and the sample separation is similar. In OOD noise (Liu et al., 2020b; Hao & Li, 2024), the criterion is usually based on the energy score:

$$h(\hat{x}_i) = E(\hat{x}_i) = -T_E \log \left\{ \mathbf{1}_C^\top \exp \left[ \frac{f_\theta(\hat{x}_i)}{T_E} \right] \right\}, \tag{2}$$

where $T_E > 0$ is the temperature parameter. Notably, the energy function measures the uncertainty of a model, where samples from the training set generally result in low energy (indicating high confidence), whereas OOD samples result in high energy (indicating low confidence). This makes energy valuable for distinguishing OOD samples (Liu et al., 2020b; Hao & Li, 2024).

## 3 METHODOLOGY

In this section, we present AEON, our one-stage training approach that simultaneously learns the ID and OOD noise rates, partitions samples into clean, ID, and OOD subsets, and optimizes the model with distinct loss functions for each subset. Hence, we aim to mitigate the issues of multistage training and sample selection processes mentioned in Section 2.2, which rely on either *(i)* non-informative clustering or *(ii)* an arbitrary (or hand-crafted) noise rate threshold. We first present the formal definitions of these label noise rates as follows:

**Definition 1** (ID label noise rate). $\eta^{id} = \Pr(\hat{y} \neq y | \hat{x})$.

**Definition 2** (OOD noise rate). $\eta^{ood} = \text{Pr}\left(y \notin \hat{\mathcal{Y}} | \hat{x}_i\right)$.

The ID and OOD noise rates defined above represent the rates of ID and OOD samples, respectively. Hence, they can be used to separate samples from the given training set in two steps: 1) sorting the criterion values and 2) thresholding at the quantile value corresponding to the noise rate. For example, if $\eta^{ood} = 0.2$, then one can sort the energy scores of all training samples and consider the 20% largest energy score samples as OOD. Two problems arise from this procedure. First, these noise rates are unknown and must be estimated. Second, the thresholding operator for separating (or selecting) samples is non-differentiable, rendering it infeasible to learn these noise rates using gradient-based approaches. In the following subsection, we relax the hard thresholding to make the sample selection process differentiable to learn these noise rates with a gradient.

### 3.1 NOISE RATE PARAMETERISATION

Since the ID and OOD sample selection processes are similar, we only explain The case for ID sample selection. The case for OOD sample selection can be performed using the same procedure.

To integrate the label noise rate $\eta^{id}$ into the thresholding operation of the small loss criterion, it must be converted from its original form (i.e., percentage) into a "loss" value. This can be done in two steps: 1) fitting a distribution, such as a normal distribution, on the losses of all training samples, denoted as $\mathcal{N}(\mu, \sigma^2)$, and 2) calculating the value at the quantile $\eta^{id}$ as follows:

$$\tau\left(\eta^{id}\right) = \Phi^{-1}\left(1 - \eta^{id} \,\middle|\, \mu, \sigma^2\right), \tag{3}$$

where $\Phi^{-1}()$ is the inverse cumulative function of normal distribution.

The conversion of the label noise rate $\eta^{id}$ into a loss-based threshold $\tau(\eta^{id})$ enables the separation of the samples into clean and noisy sets. Conventional thresholding assigns binary weights (0 or 1) to each sample, making the process non-differentiable with respect to $\eta^{id}$. To overcome this limitation, we introduced a soft thresholding mechanism, where the weight of each sample is computed using a sigmoid function as follows:

$$w_i\left(\eta^{id}\right) = \frac{1}{1 + \exp\left[-\frac{\tau(\eta^{id}) - \ell_{CE}(f_\theta(\hat{x}_i), \hat{y}_i)}{\beta}\right]}, \tag{4}$$

where $\ell_{CE}(.,.)$ is the cross-entropy (CE) loss defined in Eq. (1), and $\beta > 0$ is a temperature to adjust the stiffness of the sigmoid function. Fig. 2 (middle part) illustrates the main idea of soft-thresholding through the weight in Eq. (4).

Given the weight $w_i(\eta^{id})$ in Eq. (4), one can simply weight the contribution of sample $\hat{x}_i$ to train both the model of interest and the label noise rate $\eta^{id}$ using gradient descent. For example, the parameters $\theta$ and $\eta^{id}$ can be learned using the CE loss $\ell_{CE}(.)$ for the clean-label samples and a consistency loss $\ell_u(.)$ for the noisy-label samples (Li et al., 2020), as follows:

$$\min_{\theta, \eta^{id}} \frac{1}{N} \sum_{i=1}^{N} w_i\left(\eta^{id}\right) \ell_{CE}\left(f_\theta(\hat{x}_i), \hat{y}_i\right) + \left[1 - w_i\left(\eta^{id}\right)\right] \ell_u(\hat{x}_i, \theta). \tag{5}$$

Note that naively estimating the label noise rate $\eta^{id}$ via gradient descent may result in values for $\eta^{id}$ outside $[0, 1]$, so we parameterize $\eta^{id}$ through the following sigmoid transformation to avoid this issue:

$$\eta^{id} = \frac{1}{1 + \exp(-\gamma^{id}/T)}, \tag{6}$$

where $\gamma^{id} \in \mathbb{R}$ is a learnable parameter and $T$ is a temperature parameter. Hence, we can directly learn $\gamma^{id}$ and apply Eq. (6) to recover $\eta^{id}$. Similarly, for OOD sample selection, we apply the same procedure using energy scores as the criterion. Energy scores $E(\hat{x}_i)$ from Eq. (2) are fitted to a normal distribution $\mathcal{N}(\mu_E, \sigma_E^2)$, and the OOD threshold is computed as $\tau(\eta^{ood}) = \Phi^{-1}(1 - \eta^{ood} | \mu_E, \sigma_E^2)$. The OOD sample weights are then:

$$w_i(\eta^{ood}) = \frac{1}{1 + \exp\left[-\frac{\tau(\eta^{ood}) - E(\hat{x}_i)}{\beta}\right]}, \tag{7}$$

The computational complexity of calculating the sample weighting in Eq. (4) is $\mathcal{O}(BC)$, where $B$ is the mini-batch size and $C$ is the number of classes. Further analysis in Appendix E.

## 3.2 Learning Algorithm

Our proposed AEON learning algorithm integrates the noise rate parametrization from Section 3.1 to formulate the following three complementary objectives: *(i)* a supervised loss for samples likely to have clean labels, *(ii)* an unsupervised loss for samples suspected of ID noise, and *(iii)* a contrastive loss to improve representation learning, particularly for OOD samples. Unlike traditional multistage approaches, AEON does not assign samples exclusively to any of the objectives mentioned above. Instead, each sample adaptively contributes to all losses, weighted by its estimated likelihood of being clean, ID, or OOD through $w_i(\eta^{\mathrm{id}})$ and $w_i(\eta^{\mathrm{ood}})$. In the remainder of this paper, we drop the dependency on $\eta$ from $w$ to simplify the notation.

The overall AEON objective function for learning the parameters $\theta^*$, $\gamma^{\mathrm{id}^*}$, and $\gamma^{\mathrm{ood}^*}$ is defined as:

$$\theta^*, \gamma^{\mathrm{id}^*}, \gamma^{\mathrm{ood}^*} = \underset{\theta, \gamma^{\mathrm{id}}, \gamma^{\mathrm{ood}}}{\operatorname{argmin}} \frac{1}{N} \sum_{i=1}^{N} w_i^{\mathrm{ood}} \mathcal{L}_i^{\mathrm{id}} + \left(1 - w_i^{\mathrm{ood}}\right) \mathcal{L}_i^{\mathrm{ood}} + \lambda \mathcal{L}_i^{\mathrm{cont}}, \tag{8}$$

where $\mathcal{L}_i^{\mathrm{id}}$ and $\mathcal{L}_i^{\mathrm{ood}}$ are the in-distribution and out-of-distribution loss functions, and $\lambda \in [0, 1]$ is a hyper-parameter controlling the contribution of contrastive learning loss $\mathcal{L}_i^{\mathrm{cont}}$.

**The in-distribution loss**   is defined as:

$$\mathcal{L}_i^{\mathrm{id}} = w_i^{\mathrm{id}} \, \ell_{\mathrm{CE}} \left(f_\theta(\hat{x}_i), \hat{y}_i\right) + (1 - w_i^{\mathrm{id}}) \ell_u(\hat{x}_i, \theta) + \max(0, E(\hat{x}_i) - m^{\mathrm{id}})^2, \tag{9}$$

where $\ell_{\mathrm{CE}}(., .)$ is the cross-entropy loss defined in Eq. (1), $E(.)$ is the energy function defined in Eq. (2), $m^{\mathrm{id}}$ is a hyper-parameter representing the energy margin, and $\ell_u(., .)$ is an unsupervised consistency loss defined by

$$\ell_u(\hat{x}_i, \theta) = -(\hat{q}(\hat{x}_i))^\top \log \operatorname{softmax}(f_\theta(\hat{x}_i)), \tag{10}$$

where $\hat{q}(\hat{x}_i)$ is the average of the model predictions evaluated on multiple augmentations of the sample $\hat{x}_i$. The final term in Eq. (9) is a hinge loss added as a regularization on the energy score to penalize in-distribution samples with high energy values (Liu et al., 2020b).

**The out-of-distribution loss**   leverages the energy function, which is defined as:

$$\mathcal{L}_i^{\mathrm{ood}} = \max(0, m^{\mathrm{ood}} - E(\hat{x}_i))^2, \tag{11}$$

where $E(\hat{x}_i)$ is the energy score defined in Eq. (2)) and $m^{\mathrm{ood}}$ is a hyperparameter denoting the margin. This encourages OOD samples to maintain high energy, improving the separation between in-distribution and out-of-distribution data (Liu et al., 2020b).

**The contrastive loss**   enhances feature discrimination under noisy conditions with:

$$\mathcal{L}_i^{\mathrm{cont}} = \alpha_{\mathrm{cont}} \cdot \mathcal{L}_i^{\mathrm{cont,sup}} + (1 - \alpha_{\mathrm{cont}}) \cdot \mathcal{L}_i^{\mathrm{cont,uns}}, \tag{12}$$

where $\alpha_{\mathrm{cont}}$ balances the supervised and unsupervised contrastive terms. The supervised contrastive loss function in Eq. 12 clusters samples sharing the same label while simultaneously distancing samples with differing labels as follows:

$$\mathcal{L}_i^{\mathrm{cont,sup}} = -\log \frac{\sum_{j \in \mathcal{P}_i} \exp(\mathcal{S}_{ij})}{\sum_{k=1}^{N} \exp(\mathcal{S}_{ik})}, \tag{13}$$

where $\mathcal{P}_i = \{j | \hat{y}_j = \hat{y}_i\}$. Additionally, the unsupervised term in Eq. 12 ensures representation consistency by aligning each sample more closely with its augmented version, with

$$\mathcal{L}_i^{\mathrm{cont,uns}} = -\log \frac{\exp(\mathcal{S}_{ii})}{\sum_{k=1}^{N} \exp(\mathcal{S}_{ik})}. \tag{14}$$

In Eq. (13) and Eq. (14), the similarity $\mathcal{S}_{ij}$ is computed with

$$\mathcal{S}_{ij} = \frac{g(f_\theta(\mathcal{T}_w(\hat{x}_i)))^\top g(f_\theta(\mathcal{T}_s(\hat{x}_j)))}{T_c}, \tag{15}$$

where $g(\cdot)$ is a projection head, $\mathcal{T}_w$ and $\mathcal{T}_s$ are weak and strong augmentations, respectively, and $T_c$ is a temperature parameter. Contrastive learning improves representation robustness by pulling together semantically similar samples and pushing apart dissimilar samples, thereby mitigating the effect of noisy labels (Albert et al., 2023)[2]

## 4 INSTANCE-DEPENDENT COMBINED OPEN- AND CLOSED-SET NOISE (ID2NOISE) BENCHMARK

The ID2Noise benchmark evaluates instance-dependent label noise that combines both ID and OOD corruption on the ciFAIR-100 dataset (Barz & Denzler, 2020). Unlike prior benchmarks, which assume instance-independent noise or consider only one type, ID2Noise integrates both to better reflect the real-world conditions.

Our construction process consisted of two stages. **Stage 1 (OOD Noise):** Given the OOD noise rate defined by $r^{\text{ood}} \in \{0.2, 0.4, 0.6\}$, replace $r^{\text{ood}} \times |\mathcal{D}|$ ciFAIR-100 samples with visually similar images from Places365 (Zhou et al., 2017), selected by cosine similarity in a ResNet-18 (He et al., 2016a) feature space. This produces instance-dependent OOD corruption, where semantically similar classes (e.g., bed vs. chair) are more likely to be confused. **Stage 2 (ID Noise):** Given the target ID noise rate $r^{\text{id}} \in \{0.2, 0.4\}$, we sample instance-specific noise rates $q_i \sim \mathcal{N}(\mu = r^{\text{id}}, \sigma = 0.1)$ clipped to $[0, 1]$, then apply instance-dependent label transitions to the remaining samples using the softmax-based corruption model of (Xia et al., 2020): $P(\hat{y} \mid y = y_i, \hat{x} = x_i) = \text{softmax}(q_i \times (\hat{x}_i \cdot \mathbf{w}_{y_i}))$, where $\mathbf{w}_{y_i}$ is a class-specific parameter. This ensures that the expected noise rate across the dataset equals $r^{\text{id}}$ while introducing instance-dependent variation. This introduces realistic label ambiguity while preserving the target noise rate across $\mathcal{D}$. To assess the model performance, we measured both the classification accuracy and expected calibration error (ECE) on the test set. Fig. 4 in Appendix B illustrates this two-stage pipeline, which captures the coexistence of OOD and ID noise under realistic conditions.

## 5 EXPERIMENTS

We conducted experiments across two categories of datasets to validate the effectiveness of our method. First, we evaluate AEON on two synthetic benchmark datasets:(1) our *ID2Noise benchmark* presented in Section 4; and (2) *synthetic open-set benchmarks* (Albert et al., 2022b; 2023; 2025) built from CIFAR-100 (Krizhevsky & Hinton, 2009) with random controlled noise injection for an open set (INet32 (Deng et al., 2009) and Places365 (Zhou et al., 2017)) and random closed set noise, examining multiple noise configurations. Second, we assess AEON on real-world datasets, including *Clothing1M* (Xiao et al., 2015), containing $1M$ images from shopping websites; *mini-WebVision* (Li et al., 2020), comprising $66,000$ web-crawled images across 50 classes from WebVision (Li et al., 2017), and *WebFG-496* (Sun et al., 2021) subsets (Web Aircraft, Web Bird, and Web Car) that present challenges for open and closed-set noise. Details on the datasets, benchmarking, and implementation are provided in Appendix C. A discussion of AEON's limitations, implications, and future directions is presented in Appendix I, with Data Availability in Appendix J.

### 5.1 RESULTS

**ID2Noise Benchmark** Figure 3 demonstrates the value of estimating ID and OOD noise rates on our proposed ID2Noise benchmark. We observed that explicit noise rate estimation substantially enhanced classification accuracy compared with leading noisy-label learning approaches, such as PLS (Albert et al., 2023), which rely on confidence-based sample weighting without explicitly estimating noise rates during training. Our proposed method (AEON) achieves superior accuracy (Figure 3(left)) by precisely estimating and leveraging both the ID and OOD noise rates during training (Figure 3(right)). Moreover, we establish SOTA results on the proposed ID2Noise benchmark, as shown in Table 1. The evaluation includes classic and more recent approaches, such as DivideMix (Li et al., 2020), ELR (Liu et al., 2020a), EvidentialMix (Sachdeva et al., 2021), PLS (Albert et al., 2023), and MDM (Fooladgar et al., 2024). Under varying noise rate configurations, AEON achieves superior classification performance with margins ($\Delta_{acc}$) of

---

[2]The complete training procedure is detailed in Algorithm 1 (Appendix H).

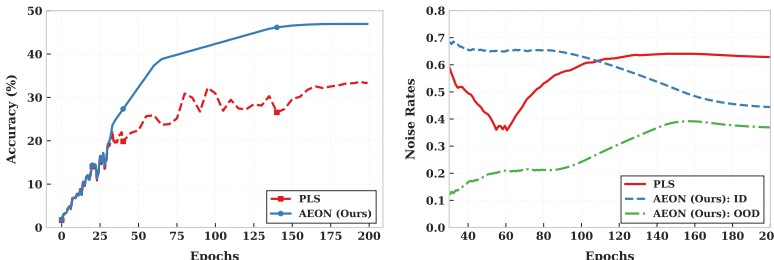

Figure 3: Correlation between the noise rate estimation ($\hat{\eta}$) and model performance on the ID2Noise benchmark with $40\%$ closed-set IDN ($r^{\text{in}}$) and $40\%$ open-set noise ($r^{\text{out}}$). (Left) Classification accuracy of AEON compared to PLS (Albert et al., 2023) over training epochs. (Right) AEON's estimation of the ID ($\hat{\eta}^{\text{in}}$) and OOD ($\hat{\eta}^{\text{out}}$) noise For PLS, post-hoc noise estimation is shown using confidence values, as it does not directly estimate noise rates.

Table 1: **Performance on ID2Noise Benchmark.** Classification accuracy (%) and ECE (%) on ID2Noise benchmark based on ciFAIR-100 (Barz & Denzler, 2020) under instance-dependent open-set noise from Places365 (Zhou et al., 2017) and instance-dependent closed-set noise from Part-Dependent (Xia et al., 2020). The first two columns ($r^{\text{ood}}$, $r^{\text{id}}$) indicate the injected open- and closed-set noise rates, respectively, whereas the last two columns show AEON's estimated noise rates ($\hat{\eta}^{\text{ood}}$, $\hat{\eta}^{\text{id}}$). All results were averaged over three runs. **Bold** numbers indicate SOTA. The statistical analysis is provided in Appendix G.

| $r^{\text{ood}}$ | $r^{\text{id}}$ | CE | DM | ELR | EDM | PLS | CECL | CrowdW | GraphNR | MDM | **AEON (Ours)** Acc./ECE ($\uparrow/\downarrow$) | $\hat{\eta}^{\text{ood}}$ | $\hat{\eta}^{\text{id}}$ |
|---|---|---|---|---|---|---|---|---|---|---|---|---|---|
| | | | | | | **ID2Noise Benchmark Results** | | | | | | | |
| 0.2 | 0.2 | 48.07/7.82 | 73.10/5.94 | 65.27/6.45 | 68.94/6.12 | 73.50/2.89 | 72.34/3.12 | 71.89/3.45 | 72.78/3.01 | 70.28/3.45 | **74.51**\*/3.24 | 0.24 | 0.31 |
| 0.4 | 0.2 | 39.52/8.45 | 63.13/6.78 | 58.32/7.12 | 54.47/7.34 | 65.25/5.67 | 66.78/4.23 | 64.45/4.89 | 66.12/4.56 | 62.90/3.82 | **69.28**\*\*/3.58 | 0.43 | 0.25 |
| 0.6 | 0.2 | 27.58/9.67 | 41.67/7.89 | 42.77/7.56 | 40.65/8.23 | 42.43/3.34 | 45.89/4.67 | 44.12/5.12 | 46.34/4.23 | 44.63/4.15 | **51.37**\*\*/3.92 | 0.57 | 0.23 |
| 0.4 | 0.4 | 21.35/10.23 | 29.54/8.45 | 29.91/8.67 | 21.40/3.12 | 33.45/4.23 | 38.67/5.01 | 36.89/5.34 | 39.12/4.78 | 37.79/4.38 | **46.94**\*\*\*/4.15 | 0.36 | 0.44 |

$^*p-value < 0.05$, $^{**}p-value < 0.01$, $^{***}p-value < 0.001$ vs. best baseline. See Appendix G for details.

$\{1.01\%, 4.03\%, 6.74\%, 9.15\%\}$ at $(r^{\text{ood}}, r^{\text{id}}) \in \{(0.2, 0.2), (0.4, 0.2), (0.6, 0.2), (0.4, 0.4)\}$ respectively. This improved performance is particularly significant, given the challenging nature of instance-dependent noise, which correlates with the image content. The ECE improvements follow a similar pattern, with AEON reducing the calibration error by $\{0.35\%, 1.09\%, 2.83\%, 3.57\%\}$ compared to the best baseline for the same noise configurations. These gains in calibration quality reinforce AEON's superior handling of instance-dependent noise.

**Synthetic Open-set Benchmarks (Albert et al., 2022b; 2023; 2025)** Table 2 presents comparative analysis on CIFAR-100 (Krizhevsky & Hinton, 2009) with controlled open-set noise from INet32 (Deng et al., 2009) and Places365 (Zhou et al., 2017). The AEON demonstrated consistent performance improvements across noise rate configurations $(r^{\text{ood}}, r^{\text{id}}) \in \{(0.2, 0.2), (0.4, 0.2), (0.6, 0.2), (0.4, 0.4)\}$, achieving average classification accuracy gains of $\Delta_{acc} = 2.25\%$ and $\Delta_{acc} = 3.1\%$ on INet32 (Deng et al., 2009) and Places365 (Zhou et al., 2017), respectively. Notably, the performance margin widened at higher noise rates (up to $3.47\%$ on INet32 and $4.12\%$ on Places365 at $(r^{\text{ood}}, r^{\text{id}}) = (0.6, 0.2)$), suggesting enhanced robustness against severe corruption. The ECE analysis revealed consistent calibration improvements, with AEON reducing the calibration error by an average of $2.15\%$ on INet32 and $2.85\%$ on Places365. The improvements were most pronounced at severe noise levels, reaching $3.27\%$ and $3.89\%$ reductions on INet32 and Places365, respectively, at $(r^{\text{ood}}, r^{\text{id}}) = (0.6, 0.2)$, which is consistent with the improvements.

**Real-world Performance Analysis** Our evaluation of large-scale real-world datasets demonstrates AEON's performance capability, where noise patterns emerge naturally from web collection processes. On Clothing1M (Xiao et al., 2015), AEON achieved $75.5\%$ Top-1 accuracy without using clean validation data during training, as shown in Table 3. The estimated noise rate aligned with previous studies (Xiao et al., 2015; Garg et al., 2024), validating our noise modelling approach. Sim-

Table 2: **Synthetic Open-set Benchmarks.** Classification accuracy (%) and ECE (%) on CIFAR-100 (Krizhevsky & Hinton, 2009) under random open-set from ImageNet32 (Deng et al., 2009) and Places365 (Zhou et al., 2017), and random closed-set noise (Albert et al., 2023). The first two columns ($r^{ood}$, $r^{id}$) indicate the injected open- and closed-set noise rates, respectively, whereas the last two columns show AEON's estimated noise rates ($\hat{\eta}^{ood}$, $\hat{\eta}^{id}$). The results were averaged over three runs. **Bold** indicates SOTA.

| $r^{ood}$ | $r^{id}$ | CE | ELR | EDM | RRL | SNCF | PLS | CECL | CrowdW | GraphNR | MDM | AEON (Ours) | | |
|---|---|---|---|---|---|---|---|---|---|---|---|---|---|---|
| | | | | | | | | | | | | Acc./ECE (↑/↓) | $\hat{\eta}^{ood}$ | $\hat{\eta}^{id}$ |
| | | | | | | | **ImageNet32 OOD Noise** | | | | | | | |
| 0.2 | 0.2 | 63.68 | 68.71 | 71.03 | 72.64 | 72.95 | 76.29 | 74.85 | 73.21 | 75.12 | 75.30 | **76.20**/3.21 | 0.23 | 0.26 |
| 0.4 | 0.2 | 58.94 | 63.21 | 61.89 | 66.04 | 67.62 | 72.06 | 70.45 | 68.89 | 71.34 | 69.10 | **74.90**/3.45 | 0.36 | 0.24 |
| 0.6 | 0.2 | 46.02 | 44.79 | 21.88 | 26.76 | 53.26 | 57.78 | 55.67 | 54.12 | 56.89 | 59.80 | **63.27**/3.82 | 0.63 | 0.23 |
| 0.4 | 0.4 | 41.39 | 34.82 | 24.15 | 31.29 | 54.04 | 56.92 | 58.45 | 57.23 | 59.67 | 63.00 | **65.78**/3.94 | 0.37 | 0.42 |
| | | | | | | | **Places365 OOD Noise** | | | | | | | |
| 0.2 | 0.2 | 59.88 | 68.58 | 70.46 | 72.62 | 71.25 | 76.35 | 75.12 | 73.89 | 74.67 | 74.50 | **77.89**/3.18 | 0.22 | 0.29 |
| 0.4 | 0.2 | 53.46 | 59.47 | 58.01 | 58.60 | 64.03 | 71.65 | 69.34 | 67.78 | 70.12 | 69.30 | **73.26**/3.52 | 0.45 | 0.27 |
| 0.6 | 0.2 | 39.55 | 37.10 | 23.95 | 49.27 | 49.83 | 57.31 | 54.89 | 53.45 | 55.67 | 59.00 | **60.74**/3.89 | 0.58 | 0.29 |
| 0.4 | 0.4 | 32.06 | 34.71 | 20.33 | 26.67 | 50.95 | 55.61 | 56.78 | 55.34 | 57.89 | **62.50** | 58.20/4.12 | 0.43 | 0.36 |

Table 3: **Real-world Dataset Performance.** (left) Performance on Clothing1M (Xiao et al., 2015) ($N = 14$, $|\mathcal{D}| = $ 1M) and mini-WebVision (Li et al., 2020) ($N = 50$, $|\mathcal{D}| = $ 66K). (right) Classification accuracy (%) on WebFG-496 (Sun et al., 2021) fine-grained recognition benchmark across aircraft ($N = 100$), bird ($N = 200$), and car ($N = 196$) categories. **Bold** numbers indicate SOTA.

| Method | Clothing1M | mini-WebVision | |
|---|---|---|---|
| | Accuracy (%) | Top-1 (%) | Top-5 (%) |
| DivideMix | 74.6 | 77.2 | 91.6 |
| ELR+ | 71.5 | 63.6 | 83.5 |
| CC-GM | 75.3 | 80.0 | 93.8 |
| MDM | 73.1 | 78.4 | 92.0 |
| CECL | 74.8 | 78.9 | 93.2 |
| CrowdW | 73.9 | 77.6 | 92.5 |
| CLIPCleaner | 73.4 | 81.6 | 93.3 |
| CLIPCleaner + DivideMix | 74.9 | **81.8** | 93.5 |
| AEON (Ours) | **75.5** | 79.8 | 94.1 |
| $\hat{\eta}^{id}$ | 0.31 | 0.18 | |
| $\hat{\eta}^{ood}$ | 0.05 | 0.17 | |

| Method | Aircraft | Bird | Car |
|---|---|---|---|
| CE | 60.80 | 64.40 | 60.60 |
| Co-teaching | 79.54 | 76.68 | 84.95 |
| PENCIL | 78.82 | 75.09 | 81.68 |
| SELFIE | 79.27 | 77.20 | 82.90 |
| DivideMix | 82.48 | 74.40 | 84.27 |
| Peer-learning | 78.64 | 75.37 | 82.48 |
| PLC | 79.24 | 76.22 | 81.87 |
| PLS | 87.58 | 79.00 | 86.27 |
| AEON (Ours) | **89.23** | **82.19** | **88.56** |
| $\hat{\eta}^{id}$ | 0.21 | 0.18 | 0.28 |
| $\hat{\eta}^{ood}$ | 0.16 | 0.11 | 0.19 |

ilarly, on mini-WebVision (Li et al., 2020), we observed competitive performance with Top-1/Top-5 accuracies of 79.8%/94.1%.

The fine-grained recognition results on WebFG-496 (Sun et al., 2021) presented in Table 3 further validate AEON's effectiveness across diverse visual domains. We achieved mean accuracy improvements $\Delta_{acc}$ of 2.29%, 3.19%, and 1.65% for the car, bird, and aircraft categories, respectively, compared with the strongest baseline model (Albert et al., 2023). Our approach yields consistent improvements in both synthetic and real-world corruptions while maintaining a strong performance without accessing clean validation data. The consistent performance improvements highlight the ability of the proposed method to handle subtle noise patterns, thereby demonstrating its strength.

An interesting observation from the real-world performance analysis presented in Table 3 is that the ranking of the results in our synthetic benchmark (see Table 1) closely mirrors those of the real-world datasets Clothing1M (Xiao et al., 2015) and mini-WebVision (Li et al., 2020). This contrasts with the rankings observed in the previously proposed benchmarks (Table 2), which show a different trend. Specifically, when focusing on the models common to both Table 1 and Table 3, our method achieved the highest performance, followed by MDM (Fooladgar et al., 2024), ELR (Liu et al., 2020a), and DM (Li et al., 2020). Conversely, the results in Table 2 present an inconclusive ranking, with MDM (Fooladgar et al., 2024) outperforming AEON. Therefore, our benchmark offers a more reliable framework for assessing the effectiveness of new methods for learning with ID and OOD instance-dependent noisy labels. It is important to note that method rankings on ID2Noise (Table 1) show strong correlation with real-world dataset performance (Table 3), which does not happen with conventional synthetic benchmarks (Table 2), validating the efficacy of ID2Noise. Further implementation details are provided in Appendix B.

**Ablation Studies** To assess component contributions and parameter sensitivity, we provide ablation studies in Appendices D–E. Our framework uses default parameters (Appendix F) and shows robust performance. Statistical analysis using t-tests and effect sizes is detailed in Appendix G.

## 6    RELATED WORK

We discuss label noise challenges, summarize noise detection and semi-supervised learning methods, and outline ID and OOD noise modelling. For details, see Appendix A.

**Real-world Label Noise** The seminal work by Xiao et al. (2015) introduced class-conditional transition matrices under instance-independent assumptions. However, empirical studies (Li et al., 2017; Wei et al., 2022; Albert et al., 2022b; Chen et al., 2024; Garg et al., 2024) show real-world datasets exhibit complex, *instance-dependent* noise patterns involving both in-distribution (ID) and out-of-distribution (OOD) samples. Synthetic benchmarks (Krizhevsky & Hinton, 2009; Xia et al., 2020; Barz & Denzler, 2020) fail to capture these intricacies, motivating our new approach.

**Evolution of Noise Detection Methods** Early approaches used the observation that deep networks learn clean samples before memorising noisy ones (Arpit et al., 2017; Liu et al., 2020a), leading to small-loss-based filtering and mixture-model strategies like DivideMix (Li et al., 2020) and PASS (Garg et al., 2023b). These methods degrade OOD-corrupted data, where bimodal loss separation fails. Feature-space-based methods (Kim et al., 2021; Albert et al., 2022a; Li et al., 2021) improved robustness but relied on multistage pipelines without explicit noise-rate estimation. Our approach addresses this issue via one-stage method that jointly estimates the ID-OOD noise rates.

**Semi-Supervised Learning with Noisy Labels** Recent studies reframe noisy-label learning as a semi-supervised problem (Liu et al., 2022a;b; 2025), treating potentially noisy labels as unlabelled data. DivideMix (Li et al., 2020) pioneered this, with extensions like ScanMix (Sachdeva et al., 2023) and PropMix (Cordeiro et al., 2021) introducing semantic clustering and proportional mixing. These methods rely on dual-model architectures and fail to address OOD-corrupted samples, motivating AEON's unified design of AEON.

**Joint ID and OOD Noise Handling** Recent studies explored simultaneous ID and OOD noise handling, such as CECL (Wan et al., 2024) and crowdsourcing methods (Nguyen et al., 2024). However, these methods rely on implicit sample filtering or require costly annotations without integrating explicit noise rate estimation. Garg et al. (2024) addressed ID noise rate estimation but did not handle OOD. AEON advances by estimating both the ID and OOD rates within a single-stage framework.

## 7    CONCLUSION

AEON represents a significant advancement in handling noisy labels for image classification, being the first method to jointly estimate both in-distribution (ID) and out-of-distribution (OOD) instance-dependent noise rates ($\hat{\eta}^{\text{id}}$, $\hat{\eta}^{\text{ood}}$) without requiring clean validation data or arbitrary thresholds. Through its novel sigmoid-based soft masking mechanism, AEON achieves SOTA performance across multiple benchmarks with reasonable computational overhead ($2.6\times$ slower than baseline CE training and $1.2\times$ slower than the previous SOTA, with linear scaling as analyzed in Appendix E) Additionally, we introduce a novel instance-dependent synthetic benchmark (ID2Noise) that better represents real-world noise, providing the community with a more realistic testbed for evaluating noise-robust methods in the future. On our proposed ID2Noise benchmark and challenging real-world datasets, such as Clothing1M and WebFG-496, AEON's ability to accurately estimate noise rates while maintaining a high classification accuracy demonstrates its practical applicability. This study opens new avenues for research on adaptive noise estimation techniques and their integration with semi-supervised learning frameworks. Our framework includes preliminary adaptive parameter scheduling capabilities (Appendix F). However, our method has several limitations that warrant future investigations. Future research directions include automatic parameter adaptation strategies (conceptual framework) discussed in Appendix F.2, establishing theoretical bounds for noise rate estimation, addressing class imbalance effects in dual-noise scenarios, and evaluating the scalability to larger datasets with thousands of classes. [3]

---

[3]Digital writing assistance tools were only used to correct grammar and and formatting. All research contributions, methodology, experimental design, and scientific insights are the original work of the authors.

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
