# OpenReview forum: "AEON: Adaptive Estimation of Instance-Dependent In-Distribution and Out-of-Distribution Label Noise for Robust Learning"
_ICLR.cc/2026/Conference — ICLR 2026 Conference Withdrawn Submission_

### Official Review · Reviewer_K8Qm · 2025-10-21

**Soundness:** 3
**Presentation:** 4
**Contribution:** 2
**Rating:** 2
**Confidence:** 5

**Summary:**

The paper investigates a practical setting of learning with label noise in which instance-dependent noise and out-of-distribution (OOD) noise coexist. To address these challenges, the authors introduce a one-stage method for robust training under label noise. They also propose a new benchmark, ID2Noise, designed to more accurately reflect real-world noise conditions.

**Strengths:**

1. The paper is clearly written and easy to follow.
2. It tackles a more challenging and realistic label-noise setting that combines instance-dependent (ID) noise and out-of-distribution (OOD) noise.
3. It introduces a benchmark dataset covering both ID and OOD label noise, enabling systematic evaluation under practical conditions.

**Weaknesses:**

1. The method is framed as addressing instance-dependent label noise (Definition 1), but the noise rate tied to Eq. (1) aggregates over all training losses; it is dataset-dependent rather than instance-dependent. As written, the approach appears instance-independent instead of instance-dependent (ID).

2. Although the paper aims to handle ID and OOD noise simultaneously, it lacks a comprehensive analysis of how these two noise types jointly affect training. Without such analysis, the contribution reads as a combination of two noise settings rather than a principled treatment of their interaction.

3. Several reported baselines are notably below the original papers. For example, ELR+ is reported at 71.5% on Clothing1M, while the original work reports 74.81%. Given ELR+ is open-sourced and widely validated, these discrepancies need careful explanation.

4. The paper asserts that both ID and OOD noise occur in real-world datasets, yet the evaluation relies on synthetic construction. An empirical study would strengthen this claim—for instance, identifying and presenting concrete OOD examples in datasets like Clothing1M or WebVision.

**Questions:**

See weaknesses

---

### Official Review · Reviewer_q2xm · 2025-10-30

**Soundness:** 3
**Presentation:** 3
**Contribution:** 2
**Rating:** 6
**Confidence:** 4

**Summary:**

The authors address two key limitations in the LNL field:

(1) the lack of unified approaches that handle both ID and OOD label noise, and

(2) the dependence of existing methods on noise-specific or manually tuned hyper-parameters.

The main idea of the paper is to design a learnable loss function that adaptively estimates the noise ratio in both ID and OOD samples through trainable parameters.
Through extensive experiments, the authors demonstrate that their proposed method achieves SOTA performance and that the estimated noise ratios closely approximate the true noise rates.

**Strengths:**

The authors propose an intuitive yet effective idea and support it with comprehensive experimental evidence.
The adaptive weights derived from the estimated noise ratios are well-designed and theoretically sound.
Furthermore, the ablation studies in the appendix provide sufficient experimental validation for the robustness and rationale behind the weighting mechanism.

**Weaknesses:**

1. Lack of theoretical justificaiton

In my opinion, the proposed loss function is suited for a well trained model.
However, it remains unclear whether this loss can reliably approximate the true noise ratios during the training process.
Specifically, I believe that when the noise rate exceeds a certain threshold, the proposed function may no longer provide a valid estimation, potentially leading to an ill-posed problem.
For instance, in a dataset with two binary attributes, whether the noise exceeds 50% can drastically affect which attribute the model learns to classify.

2. Numerous hyperparmeters

The authors should provide a more detailed sensitivity analysis for the various hyperparameters.
Although they argue that previous OOD aware LNL methods are difficult to train due to parameter dependencies, their own approach also introduces multiple tunable parameters related to noise rate estimation.
It would strengthen the paper if the authors could show, either experimentally or theoretically, that these parameters are relatively insensitive to the true noise rate or can adapt automatically.

3. Dependence on contrastive loss

In addition to the above, the method heavily relies on having a strong vision encoder.
The large performance gap between the presence and absence of the contrastive loss in the ablation study supports this point.
However, the authors provide little theoretical or analytical discussion about the role of contrastive loss in their framework.
Since the method cannot achieve SOTA performance without it (and it appears to be deeply intertwined with the proposed learning scheme) the paper would benefit from additional experiments or theoretical analysis clarifying the relationship between the contrastive loss and the main objective.

**Questions:**

I am impressed that the authors’ approach achieves the best performance on real-world datasets among recent LNL methods. If the authors can address some of my concerns listed under Weaknesses, I would be happy to evaluate this work as a sufficiently strong contribution to the community.

Minor question (you may not answer it):
In web search collected datasets (where I believe the proposed method could be particularly effective) the nature of OOD may differ from what the authors assume. Instead of encouraging the model to consider “easy” features (e.g., resolution, grayscale) as OOD, I would like the model to focus on semantic characteristics. Which hyperparameters should be adjusted, and how, to bias the method toward semantic OOD rather than low-level cues?

---

### Official Review · Reviewer_WSLq · 2025-10-31

**Soundness:** 2
**Presentation:** 3
**Contribution:** 2
**Rating:** 4
**Confidence:** 4

**Summary:**

The paper proposes AEON, a one-stage framework for learning with noisy labels that jointly models in-distribution and out-of-distribution instance-dependent noise. The method adaptively estimates noise rates without explicit sample filtering and demonstrates state-of-the-art performance on both synthetic and real-world benchmarks.

**Strengths:**

1. The experimental validation is thorough, covering multiple variables and validating the method's effectiveness against several baselines.
2. The paper's focus on a highly realistic yet rarely addressed challenge—the concurrent appearance of instance-dependent in-distribution (ID) and out-of-distribution (OOD) label noise—is particularly relevant;
3. Paper is generally well writen and easy to follow.

**Weaknesses:**

1. First, I’m a bit unclear about the distinction between the noisy-label learning with a mix of in-distribution (ID) and out-of-distribution (OOD) instance-dependent label noise discussed in this paper and what you refer to as Joint ID and OOD Noise Handling. Is the only difference the consideration of instance-dependent label noise? Furthermore, how does the proposed method specifically address instance-dependent label noise?
2. Overall, the proposed method does not present substantial methodological innovation. Most of its design still relies on distinguishing clean and noisy samples based on differences in their loss distributions. Although I understand that the new approach achieves a one-stage framework, it’s not clear why this is advantageous—what makes a one-stage approach better than a two-stage one? From my perspective, the proposed method seems to share similar motivations and basic intuitions with previous approaches, and this distinction is not clearly articulated.
3. In addition, the paper’s use of contrastive loss, in-distribution loss, and out-of-distribution loss appears to be straightforward adaptations of existing methods, lacking a principled or theoretically grounded design motivation.
4. The justification for why minimizing the overall objective in Eq (8) leads to an optimal or correct estimation of the noise parameters ($\gamma^{id}$ and $\gamma^{ood}$) is not sufficiently discussed. The paper would be strengthened by a more formal analysis or discussion on the convergence guarantees for these parameters;
5. The use of convex combinations, such as the weighting by $w_i^{ood}$ in Eq (8) and $\alpha_{cont}$ in Eq (12), lacks theoretical or analytical justification beyond empirical results. A deeper analysis is needed to explain why these specific formulations are a principled approach to balancing the respective loss components;
6. The method introduces several key hyperparameters (e.g., $\lambda$, $m^{id}$, $m^{ood}$) that seem non-trivial to tune. For instance, the margins $m^{id}$ and $m^{ood}$ are based on energy scores, which can vary significantly across different datasets, potentially hindering the method's practical applicability and generalization;
7. The contrastive loss $\mathcal{L}_i^{cont}$ in Eq (12) does not appear to differentiate between ID and OOD samples. This implies that OOD instances are also utilized in the supervised component, which seems counter-intuitive.

**Questions:**

See the weakness.

---

### Official Review · Reviewer_fyGH · 2025-11-01

**Soundness:** 2
**Presentation:** 2
**Contribution:** 3
**Rating:** 4
**Confidence:** 4

**Summary:**

This paper addresses the challenge of classification with instance-dependent in-distribution (ID) and out-of-distribution (OOD) label noise. The proposed one-stage method AEON adaptively estimates ID and OOD label noise rates, avoiding the limitations of relying on arbitrary noise rate. The framework incorporates a dual-stream soft-masking mechanism to dynamically evaluate sample reliability and a unified multi-objective training strategy combining supervised, unsupervised, and contrastive learning.

**Strengths:**

Designs a soft-masking mechanism, converts noise rates into gradient-optimizable parameters. This mechanism addresses the non-differentiability of hard-thresholding selection, and enables dynamic weight adjustment and collaborative optimization with the model.

**Weaknesses:**

1. The assumption that losses follow a normal distribution is too strong. Numerous studies have shown that losses in noisy datasets typically exhibit a similar bimodal distribution (relatively distinct separation between clean and noisy samples) instead of a normal distribution.
2. The justification only uses loss as the criterion. Other commonly used criteria in noisy label learning, such as Jensen-Shannon (JS) divergence can be explored.
3. Theoretical analysis is lacking. No proofs for noise rate estimation error bounds, model convergence, or other key theoretical guarantees are provided.
4. Comparative experiments fail to include SOTA noisy label learning and OOD detection methods from the last three years.

**Questions:**

1. The definitions of $\eta^{id}$ (Definition 1) and $\eta^{ood}$ (Definition 2) are not rigorous. Label mismatch between true and noisy labels includes both ID and OOD noise, leading to an inclusion relationship rather than mutual exclusivity.
2. Manuscript proofreading needs improvement.  In line 341, "ciFAIR-100" should be "CIFAR-100".

---

### Note · Authors · 2025-11-24

I have read and agree with the venue's withdrawal policy on behalf of myself and my co-authors.